# Phononic bath engineering of a superconducting qubit

J. M. Kitzman [1] ✉, J. R. Lane [1], C. Undershute[1], P. M. Harrington[2], N. R. Beysengulov[1], C. A. Mikolas[1], K. W. Murch [3] & J. Pollanen [1] ✉

Phonons, the ubiquitous quanta of vibrational energy, play a vital role in the performance of quantum technologies. Conversely, unintended coupling to phonons degrades qubit performance and can lead to correlated errors in superconducting qubit systems. Regardless of whether phonons play an enabling or deleterious role, they do not typically admit control over their spectral properties, nor the possibility of engineering their dissipation to be used as a resource. Here we show that coupling a superconducting qubit to a bath of piezoelectric surface acoustic wave phonons enables a novel platform for investigating open quantum systems. By shaping the loss spectrum of the qubit via the bath of lossy surface phonons, we demonstrate preparation and dynamical stabilization of superposition states through the combined effects of drive and dissipation. These experiments highlight the versatility of engineered phononic dissipation and advance the understanding of mechanical losses in superconducting qubit systems.

Vibrational or mechanical excitations naturally exist in nearly all solid-state and quantum systems. These phononic modes can take the form of well-defined excitations that can be used as a tool for enabling highly connected multi-qubit gates in ion trap architectures[1–3] as well as the generation of entangled states in systems of superconducting qubits[4,5]. When phonons take the form of a large dissipative bath, an irreversible flow of heat allows for state initialization critical to the function of laser systems[6] and the operation of optically active spin qubits[7,8]. In superconducting quantum processors, unintended coupling to spurious phonons has been shown to generate decohering quasiparticles and lead to correlated errors and degraded device performance[9–11].

Hybrid quantum systems based on the coherent coupling of two or more distinct, but interacting, systems enable the development of advanced quantum technologies[12], and investigation into the fundamental properties of complex interacting quantum degrees of freedom. Hybrid systems based on superconducting qubits, utilizing the experimental tool-kit of circuit quantum electrodynamics (cQED)[13], are a versatile platform for creating and controlling heterogeneous quantum systems and investigating their coherent dynamics[14]. Of

particular interest is the ability to leverage the intrinsically strong nonlinearity provided by the qubit to manipulate collective mechanical and acoustic degrees of freedom and explore new regimes of circuit quantum optics using GHz-frequency phonons. By engineering strong interactions between superconducting qubits and mechanical resonators it is possible to study the quantum limits of high-frequency sound in a wide variety of systems composed of qubits coupled to bulk phonons[15,16], Rayleigh-like surface waves[17–23], as well as flexural modes in suspended structures[24–26]. Impressive experimental results have been demonstrated using integrated quantum acoustic systems, including single phonon splitting of the qubit spectrum[27–29], Wigner function negativity of an acoustic resonator[15,20,29,30], electromagnetically induced acoustic transparency[23], and phonon-mediated state transfer[4,5].

Hybrid quantum acoustics systems that integrate superconducting qubits with phononic degrees of freedom typically operate in a domain where the interaction strength between the two far exceeds the loss rate of either system. In this strong coupling regime, the emphasis is on the coherent dynamics of the coupled systems rather than on the dissipation presented to the qubit via the phononic

[1]Department of Physics and Astronomy, Michigan State University, East Lansing, MI 48824, USA. [2]Research Laboratory of Electronics, Massachusetts Institute of Technology, Cambridge, MA 02139, USA. [3]Department of Physics, Washington University, St. Louis, MO 63130, USA. ✉e-mail: kitzmanj@msu.edu; pollanen@msu.edu

bath. However, the ability to create open quantum acoustic systems in which the qubit is variably coupled to multiple mechanical degrees of freedom with vastly differing strengths and loss rates opens the door for dissipative state preparation and dynamically stabilized states in the presence of a strong drive and customized phononic loss channels. Piezoelectric surface acoustic wave (SAW) devices, which can be engineered into compact devices with sharp spectral responses[18–20], are a promising avenue for engineering highly frequency-dependent phononic dissipation channels for quantum bath engineering protocols, in which the level of surface wave dissipation and qubit coupling can be precisely designed and controlled (see Fig. 1a).

## Results and discussion

We implement a quantum acoustic bath engineering protocol using a hybrid quantum system consisting of a flux-tunable transmon qubit coupled to a SAW Fabry Pérot cavity fabricated on the surface of single-crystal lithium niobate (see Fig. 1 and "Methods"). The complex admittance that describes the electromechanical properties of the SAW device, and tailors the coupling to the phonon bath, are calculated using the coupling-of-modes method[31,32] (see Supplementary Note 1). The qubit and SAW resonator are fabricated on separate substrates and their purely capacitive coupling is mediated in a flip-chip geometry via antenna pads attached to each device in the form of a pair of parallel-plate capacitors (see Fig. 1 and Supplementary Note 1). For control and readout, the composite flip-chip device is free-space coupled to the fundamental mode of a three-dimensional (3D) electromagnetic cavity with a frequency of $\omega_c/(2\pi) = 4.788$ GHz.

As shown in Fig. 2a, we design the SAW resonator spectral response in order to access both its coherent coupling to the transmon as well as the dissipative qubit–phonon dynamics. The SAW resonator is engineered to confine a single acoustic mode, which appears as a sharp peak in the conductance of the resonator near 4.46 GHz (see Fig. 2a). Near this confined acoustic mode, we fit the simulated conductance to a Lorentzian function and extract the SAW energy decay

rate $\gamma_{SAW}/(2\pi) = 0.6$ MHz from the full width at half maximum (see Supplementary Note 1). On either side of this main SAW resonance, phononic energy loss is governed by a continuum of dissipative SAW states that manifest as frequency ripples in the effective conductance of the acoustic resonator and correspond to the leakage of surface phonons out of the SAW resonator through the acoustic Bragg mirrors (see Fig. 1b). Measurements shown in Fig. 2b reveal how the features of the SAW resonator response are imparted onto the spectroscopic properties of the hybrid system and enable access to multiple regimes of circuit quantum acoustodynamics (cQAD)[19]. The qubit and the confined SAW mode interact with a coupling rate $g_m/(2\pi) = 12 \pm 0.6$ MHz, larger than the loss rate of either system ($\gamma_{SAW}/(2\pi) = 0.6$ MHz and $\gamma_q/(2\pi) = 2.67$ MHz), which is a hallmark of the quantum acoustic strong coupling regime. Strong coupling between a resonant SAW mode and a transmon qubit has been observed in previous experiments[20,21,27], and this regime serves to demonstrate hybridization between the qubit and SAW modes. More importantly, the spectroscopy shown in Fig. 2b, c also reveals signatures of controlled surface phonon loss arising from the interaction of the qubit with the continuum of SAW modes on either side of the confined SAW mode. This interaction manifests as a series of dark states in the qubit spectrum, which appear as horizontal fringes in Fig. 2b, and corresponds to the acoustic analog of the bad-cavity limit of cQED[13]. In this dissipative regime, the dynamics of the hybrid system is dominated by the loss of phonons from the resonator that have a frequency outside of the acoustic mirror stop band (see Supplementary Note 1).

To utilize the frequency-dependent acoustic loss for dynamical quantum state stabilization, we consider the effect of phononic decay on qubit decoherence in the presence of a strong coherent drive of amplitude $\Omega$ near-resonant with the qubit. In this regime, the emission spectrum the system consists of a peak at the drive frequency $\omega_d$ and two additional sidebands at $\omega_d \pm \Omega_R$, where $\Omega_R = \sqrt{\Omega^2 + \Delta^2}$ is the generalized Rabi frequency and $\Delta = \omega_d - \omega_q$ is the detuning between the drive and the qubit[33–35] (see Fig. 3a). In the presence of a

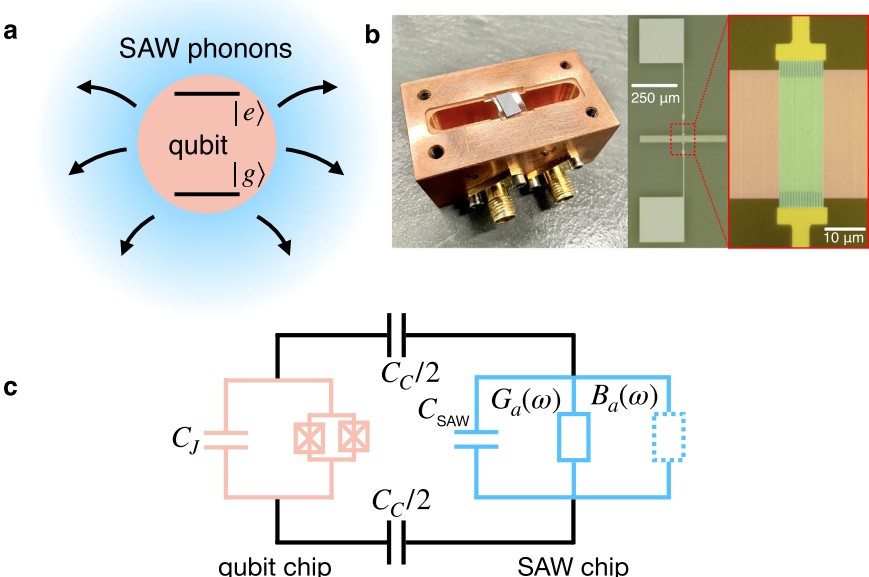

**Fig. 1 | Open quantum phononics based on engineered SAW–qubit interaction.** **a** Schematic representing a qubit coupled to a bath of SAW phonons. The qubit (salmon) nonunitarily radiates excitations into a bath of SAW phonons (blue), where the emission rate is mediated by the electrical conductance of the SAW structure. **b** Image of the flip-chip qubit–SAW hybrid device mounted in a 3D microwave cavity (left), which is used for control and readout. False color optical micrograph of the acoustic resonator (right), consisting of an IDT (green), as well as acoustic Bragg mirrors (red). **c** Equivalent circuit model of the composite qubit–SAW system. Here $C_J$ is the total capacitance shunting the Josephson junctions, $C_C$ is the capacitance responsible for coupling the qubit and SAW resonator, which is primarily dictated by the parallel-plate capacitance between the two substrates, and $C_{SAW}$ is the geometric capacitance of the SAW resonator. The complex admittance that represents the electro-mechanic response of the SAW resonator is divided into a conductance $G_a(\omega)$, as well as a susceptance $B_a(\omega)$ (see Supplementary Note 1).

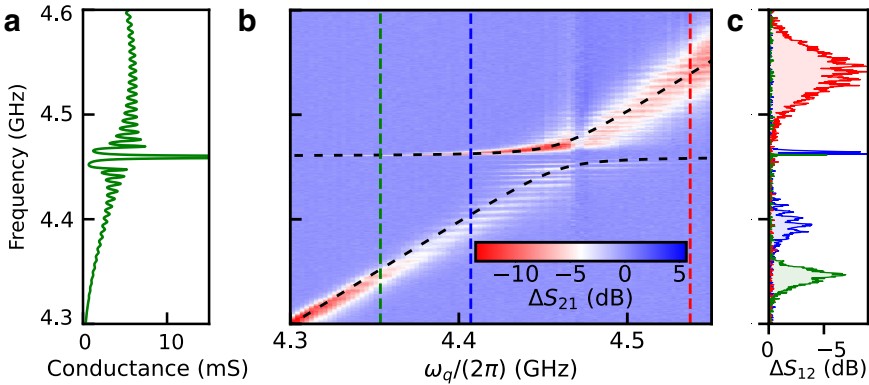

**Fig. 2 | Spectroscopic characterization of the hybrid system. a** Simulation of the SAW resonator conductance $G_a(\omega)$ based on the coupling-of-modes modeling (see Supplementary Note 1). The resonator is designed to host a single confined SAW mode, which corresponds to the narrow peak in the device conductance at 4.46 GHz, and the effective electrical conductance of the device mediates the coupling between the qubit and a given SAW mode. **b** Two-tone spectroscopy of the SAW–qubit hybrid system revealing an avoided crossing between the qubit and main SAW mode. Black dashed line: fit to the data, indicating an acoustic coupling of $g_m/(2\pi) = 12 \pm 0.6$ MHz. **c** Spectroscopy linecuts from (**b**) at the positions of the vertical dashed lines. The oscillations in frequency in each scan highlight the phononic loss channel imparted on the qubit, which arise from the modulation of the conductance of the SAW resonator and are associated with the leakage of SAW excitations through the acoustic Bragg mirrors.

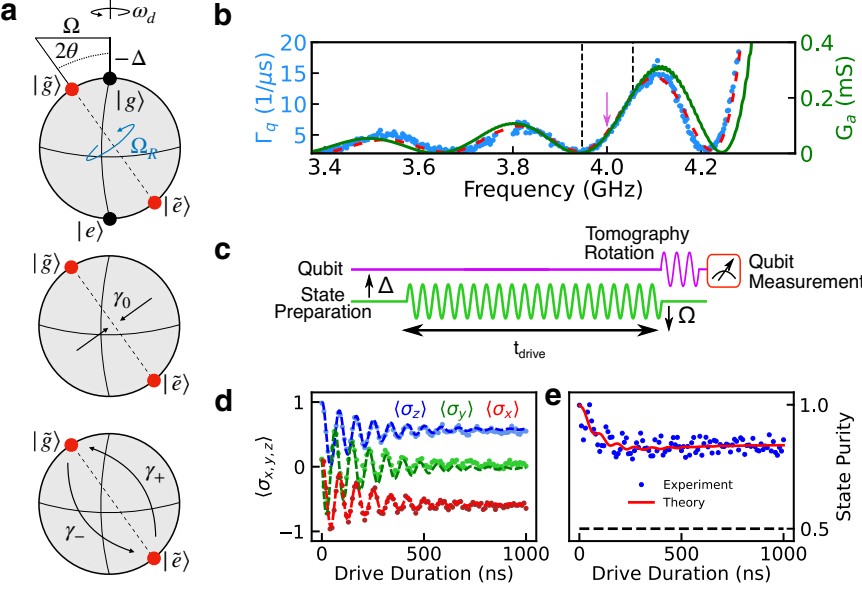

**Fig. 3 | Experimental protocol for implementing dynamical state preparation and stabilization of the driven-dissipative system. a** Top: schematic representing the rotation of the qubit eigenbasis in the presence of a detuned drive applied to the qubit. Center: representation of pure dephasing in the dressed basis. Bottom: representation of competing decay rates in the dressed qubit basis. By tailoring the coupling of the qubit to the frequency-dependent phonon bath, we are able to control the relative size of $\gamma_\pm$ as a function of qubit frequency. **b** Measurement of qubit loss $\Gamma_q = 1/T_1$ versus frequency (blue). The red curve is a fit to Eq. (1) showing that the variation in the qubit loss is dictated by conversion into SAW phonons. The green curve is a result of the coupling-of-modes model for the electrical conductance of the SAW resonator with no fit parameters (see Supplementary Note 1).

The arrow indicates the frequency at which the bath engineering experiments are performed. The black dashed lines are the endpoints of experimentally accessible Mollow triplet sidebands. **c** Pulse sequence for investigating the coherence of the driven-dissipative quantum acoustics system. **d** Tomographic reconstruction of qubit state evolution at a resonant Rabi frequency of $\Omega/(2\pi) = 8.47$ MHz and drive detuning $\Delta/(2\pi) = -10$ MHz. Dots represent the experimental data while the dashed lines are solutions to Eq. (2) with the same drive parameters, which correspond to $\gamma_+ = 3.4\ \mu s^{-1}$ and $\gamma_- = 1.3\ \mu s^{-1}$. **e** Measurement of the state purity as a function of time. In the combined presence of phonon loss and drive the purity reaches a value of 0.85 at $t = 1\ \mu s$, in contrast to a maximally mixed state represented by the dashed line at $\mathcal{P} = 0.5$. The solid red line is the expected state purity based on Eq. (2).

frequency-dependent emission spectrum, one sideband can be suppressed, leading to preferential emission from the other sideband and non-zero qubit coherence in the undriven basis for times long compared to the intrinsic lifetime of the qubit[36].

For this type of bath engineering protocol to work in our device, the SAW admittance must modify the decoherence rate of the qubit over frequencies comparable to experimentally accessible values of $\Omega_R$. To verify this, we measured the qubit decay rate $\Gamma_q = 1/T_1$ across a broad range of frequencies far-detuned from the main acoustic

resonance (see Fig. 3b) where phonon leakage through the acoustic mirrors is maximized. At these frequencies, the conductance of the SAW resonator is well approximated by that of the acoustic transducer, as the reflectivity of the acoustic Bragg mirrors is small, and the total loss of the qubit can be approximated as:

$$\Gamma_q(\omega_q) = \frac{\omega_q}{2\pi Q_i} + \Gamma_0\, \text{sinc}^2\left(\pi N_p \frac{\omega_q - \omega_s}{\omega_s}\right), \tag{1}$$

where $Q_i = 1.67 \times 10^3$ is the qubit internal quality factor, $\Gamma_0 = 0.252\,\text{ns}^{-1}$ is the maximum conversion rate of the qubit excitation into SAW phonons, $N_p = 16$ is the number of finger pairs in the IDT structure of the SAW resonator, and $\omega_s/(2\pi) = 4.504\,\text{GHz}$ is the central SAW frequency, which is within 1% of the value predicted from the device fabrication parameters (see Supplementary Note 1). The first term in Eq. (1) describes the decay of the bare qubit while the second term is associated with qubit energy conversion into SAW phonons. At $\omega_q/(2\pi) = 4.001\,\text{GHz}$, where the gradient of qubit loss into SAW phonons is large, the qubit decay rate varies by a factor of 3.7 over a span of 80 MHz ($\approx 2\Omega_R$) and allows us to use SAW-phonon modes for efficient state preparation.

The dynamics of the reduced qubit density matrix in the combined presence of the drive and frequency-dependent SAW loss are described by the Lindblad master equation[36–38]:

$$\dot{\rho} = i[\rho, H] + \gamma_0 \cos^2(\theta)\sin^2(\theta)\mathcal{D}[\tilde{\sigma}_z]\rho + \gamma_- \sin^4(\theta)\mathcal{D}[\tilde{\sigma}_+]\rho$$
$$+ \gamma_+ \cos^4(\theta)\mathcal{D}[\tilde{\sigma}_-]\rho + \gamma_1\mathcal{D}[\sigma_-]\rho + \frac{\gamma_\phi}{2}\mathcal{D}[\sigma_z]\rho, \tag{2}$$

where $\mathcal{D}[A]\rho = (2A\rho A^\dagger - A^\dagger A\rho - \rho A^\dagger A)/2$. The angle $\theta$ is defined by $\tan 2\theta = -\Omega/\Delta$ and represents the rotation of the qubit eigenbasis under the drive (see Fig. 3a and Supplementary Note 2). The operators $\tilde{\sigma}_\pm$ and $\tilde{\sigma}_z$ along with the correponding rates $\gamma_\pm$ and $\gamma_0$ represent transitions between eigenstates and dephasing in the rotated frame. Dissipation in the lab frame is represented by the operators $\sigma_-$ and $\sigma_z$ along with the rates $\gamma_1$ and $\gamma_\phi$, for qubit depolarization and dephasing. The transition rates $\gamma_\pm$ represent competing decay of the qubit into SAW phonons in the rotated basis, and by tailoring the frequency-dependent phonon bath, these rates vary significantly over the frequency scale $2\Omega_R$ as seen in Fig. 3b. In the limit $\gamma_\pm \gg \gamma_\mp$, the spectral weight of one sideband of the qubit emission spectrum is suppressed, leading to dynamical stabilization of a rotating-frame eigenstate. In general, the plane accessible within the Bloch sphere is controlled by the phase of the drive signal. In the measurements described below, we set the phase of the drive such that the qubit eigenstates lie in the $XZ$-plane of the Bloch sphere. The chosen Rabi frequency and drive detuning further constrain the qubit eigenstates to a particular axis in this plane.

To demonstrate the phononic bath engineering protocol, we flux bias the qubit to $\omega_q/(2\pi) = 4.001\,\text{GHz}$, where the gradient of the qubit loss varies strongly as a function of frequency (see Fig. 3b). By tailoring the drive parameters, the qubit emits radiation at frequencies corresponding to Mollow triplet sidebands[33] at rates governed by the SAW-phonon-induced loss. To calibrate the drive strength, we measure resonant Rabi oscillations as a function of drive amplitude and interpolate the results to obtain a mapping between drive amplitude to Rabi frequency (see Supplementary Note 3). By driving the qubit at a detuning $\Delta/(2\pi) = -10\,\text{MHz}$ with strength $\Omega/(2\pi) = 8.47\,\text{MHz}$ we demonstrate the ability to prepare a qubit state in the $XZ$-plane of the Bloch sphere, which we verify using tomographic reconstruction of the qubit state as a function of driving time $t_{\text{drive}}$ (see Fig. 3c, d). As shown in Fig. 3d, in the limit $t_{\text{drive}} \gg T_1$, the qubit density matrix approaches a fixed point determined by the drive parameters and asymmetric phononic loss. We further quantify this dissipation-enabled dynamical stabilization protocol by calculating the state purity of the qubit, $\mathcal{P} = \text{Tr}(\rho^2)$, as a function of time as shown in Fig. 3e. We see that, in the limit $t_{\text{drive}} \gg T_1$, the state purity reaches $\mathcal{P} = 0.85$, well above the value $\mathcal{P} = 0.5$ of a maximally mixed state. By including fit parameters that describe the global dephasing rate $\gamma_\phi = 1.48\,\mu\text{s}^{-1}$ and global depolarization rate of $\gamma_1 = 2.46\,\mu\text{s}^{-1}$ in the numerical solutions to Eq. (2), we find quantitative agreement with the tomography data shown in Fig. 3d, e. We note that these optimal fit parameters are in reasonable agreement with the measured values for the bare qubit decay at this frequency

$\gamma_q/(2\pi) = f_q/Q_i \simeq 2.4\,\mu\text{s}^{-1}$ and the pure dephasing rate $\gamma_{\phi,\text{exp}} \simeq 0.93\,\mu\text{s}^{-1}$ (see Supplementary Note 3).

In the basis dressed by the drive, this dynamically stabilized qubit state is effectively cooled toward thermal equilibrium via the frequency-dependent emission of energy into SAW phonons. Because the qubit-reduced density matrix is no longer evolving in time for sufficiently long values of $t_{\text{drive}}$, the driven-dissipative system has reached its steady state and the notion of an effective qubit temperature is well-defined. To quantify the efficiency of this phonon-induced qubit cooling, we consider the driven qubit as a two-level system subject to the Hamiltonian $H = \frac{\hbar\Omega_R}{2}\tilde{\sigma}_z$, in equilibrium with an effective thermal bath, where $\tilde{\sigma}_z = \sin(2\theta)\sigma_x - \cos(2\theta)\sigma_z$. The partition function for this system is $\mathcal{Z} = 2\cosh(\frac{\hbar\Omega_R\beta_{\text{eff}}}{2})$, with $\beta_{\text{eff}} = 1/k_b T_{\text{eff}}$, where $k_b$ is the Boltzmann constant and $T_{\text{eff}}$ is the effective qubit temperature. Over many repeated measurements, the average qubit energy is given by $\langle\epsilon\rangle = \frac{\hbar\Omega_R}{2}\langle\tilde{\sigma}_z\rangle$, which can used along with the partition function to relate the measured value of $\langle\tilde{\sigma}_z\rangle$ to the effective temperature $T_{\text{eff}} = -\hbar\Omega_R/(2k_b\tanh^{-1}\langle\tilde{\sigma}_z\rangle)$. With the qubit in thermal equilibrium with the surface phonon bath, we find an effective temperature of $T_{\text{eff}} \approx 250\,\mu\text{K}$ based on the experimental data in Fig. 3d, which is also the lowest effective temperature we were able to obtain in our experiments. This low effective temperature arises from the fact that the driven-dissipative protocol creates and cools an effective quantum two-level system having energy splitting $\Omega_R/(2\pi) \approx 13\,\text{MHz}$, which is significantly lower than the frequency of the qubit in the lab frame. This combination of effective temperature and transition frequency $\Omega_R/(2\pi)$ corresponds to an excited state population of ~10%, equivalent to a transmon qubit with a transition frequency of 4 GHz at a temperature of 90 mK. Finally we note that the efficiency of the driven-dissipative cooling in the current experiment is primarily limited by the relatively large rates $\gamma_1$ and $\gamma_\phi$ of the qubit. In particular, solutions to Eq. (2) predict an effective temperature as low as 85 μK if the global energy decay rate and dephasing of the qubit were improved by approximately an order of magnitude to $\gamma_1 = \gamma_\phi = 0.1\,\mu\text{s}^{-1}$.

In the combined presence of drive and phonon loss through the mirrors, the steady state of the qubit should exhibit coherence on timescales long in comparison to the intrinsic qubit lifetime in the lab frame. To investigate this steady-state behavior, we apply a coherent drive to the qubit for a time $t_{\text{drive}} = 3\,\mu\text{s}$, which is approximately one order of magnitude longer than the measured depolarization time of the qubit in the absence of drive. By choosing the parameters of the drive, we control both the rotation of the qubit eigenstates as well as the splitting of the Mollow triplet sidebands, which modifies their coupling to the lossy phononic bath. We begin by tomographically reconstructing the steady-state expectation value $\langle\sigma_x\rangle = \text{Tr}(\rho\sigma_x)$ in the bare qubit eigenbasis as a function of drive parameters and compare with the solutions to Eq. (2) as shown in Fig. 4.

As we modify parameters of the drive, we observe excellent agreement between the measured and predicted value of $\langle\sigma_x\rangle$ of the resulting qubit state, and reveal a region of zero coherence where the competing loss rates $\gamma_\pm$ in Eq. (2) cancel each other and $\langle\sigma_x\rangle = 0$. Furthermore, full tomographic measurement of the qubit state vector allows us to reconstruct the qubit-reduced density matrix (see Supplementary Note 4) and calculate the steady-state purity of the qubit as function of drive parameters as shown in Fig. 5a, which we also find to be in good agreement with solutions to Eq. (2) (see Fig. 5b). As shown in Fig. 5c, we further investigate the versatility of this state preparation protocol by plotting the purity as a function of the coordinates of the qubit state vector in the $XZ$-plane over the experimentally accessible values of the drive parameters. We find that the dissipation-enabled state preparation can be used to create high-purity superposition states across a relatively large range of $\langle\sigma_x\rangle$. However, access to large negative values of $\langle\sigma_z\rangle$ is limited in this device by the difference between $\gamma_+$ and $\gamma_-$ arising from the slope of the SAW device conductance versus frequency. We find that these experimental results are

also in good agreement with the open quantum systems modeling based on solutions to Eq. (2), which is also displayed in Fig. 5c. We note that the systematic difference in the span between the experimentally and numerically obtained values of tomography components in the *XZ*-plane likely arises from long timescale variation in the qubit frequency during the relatively long duration (several hour) tomography measurements (see Supplementary Note 4). Finally we note that in the experiments presented above, all applied signals are far-detuned from the confined resonance of the SAW device, and we therefore expect the SAW to remain near its ground state.

In conclusion, we have demonstrated a phononic open quantum system composed of a superconducting qubit coupled to an engineered bath of lossy surface acoustic wave phonons. This system enables the investigation of the dynamical and steady-state of superpositions of the qubit when it is subjected to the combined effects of strong driving and phononic loss. The lossy SAW environment allows for high-purity qubit state preparation and dynamical stabilization within a plane in the Bloch sphere. Modest improvements to the transducer design and qubit quality factor offer an avenue to prepare states with purity exceeding 99%. In particular, this could be achieved by improving the bare qubit lifetime and pure dephasing rate by order of magnitude, which would enhance the decay of qubit energy into phonons rather than the electromagnetic environment. Similarly, modifying the SAW transducer design to have a sharper spectral response would increase the relative difference of the rates $\gamma_\pm$, improving the efficiency of the phononic bath engineering protocol. Finally, we note that these results also open the door to investigating the non-unitary evolution of quantum states and effective non-Hermitian Hamiltonians hosting decoherence-induced exceptional points in open quantum acoustic systems via post-selection protocols[39].

## Methods

### Qubit fabrication

The qubit was fabricated on high resistivity silicon of thickness 275 μm using the Dolan bridge technique using a bilayer of MMA/EL9 and PMMA/C2 resists. Two Josephson junctions fabricated in parallel formed a SQUID loop with an area 16 μm². Base exposure doses for the formation of the Josephson junctions were 300 μC/cm² with an additional dose of 50 μC/cm² in order to form an undercut to ensure good liftoff. The superconducting wire was wound 30 times around the copper cavity into which the qubit was mounted, corresponding to one flux quantum threading the SQUID loop at $I_{wire} \simeq 125$ mA.

### SAW fabrication

The SAW resonator was fabricated on single-crystal YZ-cut lithium niobate of thickness 500 μm. We use a single-layer PMMA/C2 resist spun at 4000 RPM for 45 s, which is then baked at 180 °C and covered with a 30-nm aluminum discharging layer prior to electron beam lithography. Regions on the substrate with low spatial density of features are exposed with a dose of 325 μC/cm², while areas with higher density of features are exposed with a lower dose of 275 μC/cm² to account for a lower proximity dose due to forward scattering of incident electrons. After exposure, the aluminum discharging layer is removed by submerging the sample in AZ 300 MIF developer for

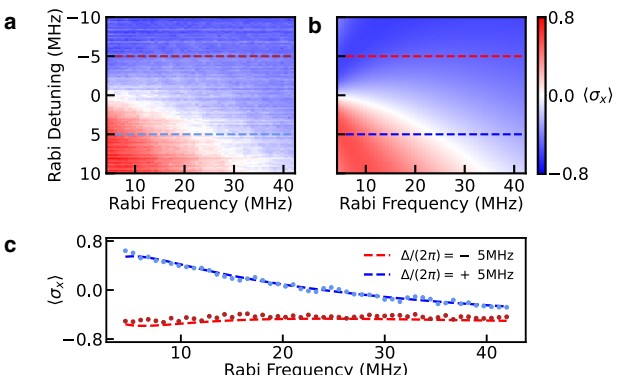

**Fig. 4 | Dissipation-enabled dynamical stabilization of the x-component of the qubit state vector. a** By varying both the strength (resonant Rabi frequency) and the detuning of the state preparation pulse relative to the qubit frequency, we are able to demonstrate dynamical state stabilization that has both positive and negative values of $\langle\sigma_x\rangle$. **b** Numerical solutions to Eq. (2) over the same range of drive parameters as in (**a**). The rates $\gamma_1$ and $\gamma_\phi$ are fit parameters as described in the text, while all other model parameters are determined empirically. **c** Representative horizontal linecuts from (**a**) along with the corresponding predictions based on the solution to the Lindblad master equation (**b**).

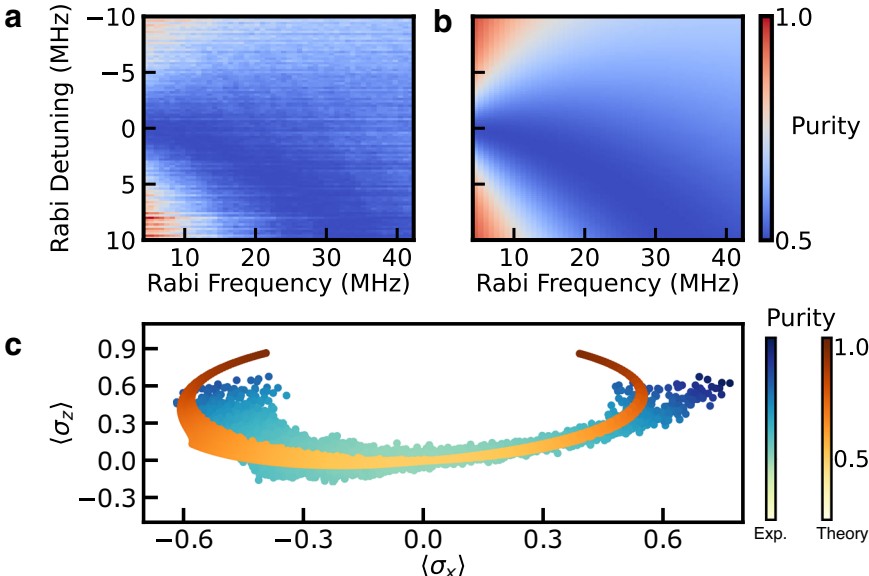

**Fig. 5 | Reconstruction of the steady-state purity as a function of the drive parameters and the coordinates in the *XZ*-plane of the Bloch sphere. a** Measured steady-state purity with the same drive parameters as in Fig. 4. **b** Purity based on solutions to Eq. (2) under same drive parameters. **c** Purity of the qubit steady state in the *XZ*-plane of the Bloch sphere across the same range of drive parameters, where solutions to Eq. (2) are overlaid on the experimental data.

200 s. The exposed resist is then developed in a solution of 1:3 MIBK-IPA solution for 50 s followed by 15 s in IPA. After development, a 30-nm aluminum layer is thermally evaporated onto the substrate to form the SAW resonator structure, followed by a lift-off procedure.

## Hybrid system assembly

Spacers patterned on the SAW chip of 4-μm-hick S1813 photoresist nominally dictate the spacing between the qubit chip and the SAW chip. These spacers are formed by spinning four individual layers of resist at 5000 RPM for 50 s each, followed by baking the resist at 110 °C following each layer. The spacers are then patterned via standard photolighographic techniques. To make the spacers robust, the resist is then hard-baked at 250 °C for 1.5 h. The $250 \times 250$ μm antenna pads on both the qubit and SAW chip are then aligned using a standard mask aligner and glued together using additional S1813 resist. Based on the coupling strength of $g_m/(2\pi) = 12 \pm 0.6$ MHz, we calculate the coupling capacitance between the qubit and SAW resonator to be 31.7 fF[40]. This value is slightly smaller than expected, indicating that the chips are farther apart than the thickness of the resist spacers, suggesting that the resist used for glue is dominating the inter-chip spacing[32].

## Data availability

The data that support the findings of this study are available at https://zenodo.org/record/8043749.

## Code availability

The analysis code that support the findings of this study are available at https://zenodo.org/record/8043749.

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

## Acknowledgements

We thank M.I. Dykman, A. Schleusner, D. Kowsari, and L. Zhang for valuable discussions. We also thank R. Loloee and B. Bi for technical

assistance and use of the W.M. Keck Microfabrication Facility at MSU. The Michigan State portion of this work was supported by the National Science Foundation (NSF) via grant number ECCS-2142846 (CAREER) and the Cowen Family Endowment at MSU. C.A.M. acknowledges support from the NSF via grant number DMR-2003815. The Washington University portion of this work was supported by the NSF via grant number PHY-1752844 (CAREER).

## Author contributions

J.M.K. and C.U. performed the experiments and analyzed the data with assistance from N.R.B. and K.W.M. J.R.L., J.M.K., and C.A.M. designed and fabricated the devices. J.M.K. and P.M.H. developed the software for numerically solving the master equation. J.P. supervised and assisted the project and provided guidance. All authors contributed to discussions and the production of the manuscript.

## Competing interests

The authors declare no competing interests.
