## [Peer Review File · Nature Communications]

REVIEWER COMMENTS

Reviewer #1 (Remarks to the Author):

The manuscript from J. M. Kitzman, et al. reports on the experimental investigation of a two-level superposition state creation and stabilization via a continuous microwave drive and an engineered dissipative phononic channel. The experiment joins other researches on hybrid quantum system between superconducting qubit, here of the transmon type, and a phononic bath, here a SAW type. It also joins another field of research on creation and stabilization of deterministic quantum state via bath engineering. The main results of the manuscript are shown in Fig.3 and Fig.4 where the authors present the creation and stabilization of a superposition state of their qubit.

According to the abstract, the authors claim:

1. to have achieved “a new type of quantum control over superconducting circuits”.
2. to be able to “prepare and stabilize arbitrary qubit states”
3. to have “find that the presence of the energy-dependent loss imparted onto the qubit by the phonons is well described by a master equation treatment of the composite system”.

I do not understand the claim 1. The ref. [37], from the same group it seems, already demonstrated the same experiment with engineered photonic loss channel. Is the fact that the loss channel is here phononic deserve such a claim? I would say no since ref. [37] already demonstrated it. This manuscript used the same “type” of quantum control on a different implementation.

I do think the claim 2 to be unsupported by the presented data and is strongly misleading on the achieved results of the experiment. Even the author themselves wrote later in the manuscript to be able to create “arbitrary qubit state in the XZ-plan” leaving a 3rd dimension of the Bloch sphere unexplored. Even worse, the data presented showed the ability to explore the X axis of the Bloch sphere without any data on the Z axis. This is surprising since a previous paper from the same group, ref. [37] showed very similar results but with the data on the Z axis. I currently think that the authors overclaim their ability to create and stabilize and “arbitrary state” and that such claim should be retracted. However I think the author are right to claim such ability in the XZ plan of the Bloch sphere but they should add data about the Z projection to support that claim.

The claim 3 is surely true for the author but is not new for the community since ref. [21] published in 2018 already saw the conformity between theory and experiments on such hybrid system by using the Lindblad master equation. Such sentence should then be removed.

To conclude, the reported experiment is a repetition of a previous experiment of the same group, ref. [37], the difference being the replacement of an engineered photonic loss channel by a photonic one. While this in itself is an achievement and is interesting, the manuscript is strongly spoiled by various unfounded claims. In its current state, this manuscript should not be published in any

journal. I do however think that the experiment in itself is of great interest should be published without the current overclaims and lack of data.

In the following, I give various remarks and question for a possible resubmission:

1. In the abstract, the author claim to be “able to prepare and stabilize arbitrary qubit states”. It is my opinion that this claim is currently not supported by the presented results and therefore should either be retracted or additional data must be presented. Indeed, the main text present the preparation, not stabilization, of arbitrary states in the XZ plane only, see Fig. 3c. Moreover even such a claim is unsupported by their data, see another of my comment later, point 7.

2. On Fig 1.c the author depicted the SAW device, in blue, to have one capacitive coupling, in black, with the qubit, in salmon. When I look at the device picture shown on Fig. 1.b I see two capacitive coupling pads and so I would have expected the same in Fig. 1.c. Can the author comment on this?

3. On the top paragraph of page 5, the authors claim “The qubit and the confined SAW mode interact with a coupling rate $gm/(2\pi) = 12 \pm 0.6$ MHz, larger than the loss rate of either system, which is a hallmark of the quantum acoustic strong coupling regime.”. There is however, at that point of the manuscript, no demonstration of the qubit lifetime nor the SAW cavity lifetime and while I agree that an avoided level crossing is a visual proof of the strong coupling regime a quantitative argument is still required to make such a claim.

It is possible to find the qubit decay rate later on at the end of page 7. There, the author discussed the “phononic bath engineering protocol” and cite the “global depolarization rate” to be about $2.565 \mu\text{s}^{-1}$, which I assume to be equivalent to a decay rate $2.565/(2\pi) = 0.41$ MHz.

I was unable to find the cavity decay rate in neither the main text nor the supplementary information. It may be possible to deduce the cavity loss rate from the supplementary material where the “SAW propagation loss” η and “speed of sound” v_{sound} is given by assuming an exponential decay behavior (expected for a SAW cavity mode) $\gamma_{\text{SAW}}/(2\pi) = 1/(\eta v_{\text{sound}}) = 1.82$ MHz.

Thus while I was, maybe, able to back-up the claim of the author I think that:

a. The qubit and cavity decay rate should be made clearer to allow an easy comparison with the coupling rate and consequently ensure the achievement of the strong coupling regime.

b. Such rate being measured, I would add such measurements in the supplementary.

4. On the top paragraph of page 5, the author claimed to have reached the strong coupling regime between the qubit and the SAW cavity. However, I fail to see the relevance of such a result in this manuscript. Indeed the demonstration of the strong coupling regime is not the main message of the paper and has already been demonstrated in ref [21] on a very similar system. Moreover the bath engineering described by the author does not seem to require a cavity, which is here only to offer a large decay rate gradient in a narrow frequency range outside of its bandwidth,

5. In the middle of page 7, the author explained that they chose $\omega_q/(2\pi) = 4.001$ GHz. I do think that such frequency was chosen so that γ_{-1} and γ_{-} is minimum while γ_{+} is maximum. I would

a. Add the value of γ_+ and γ_- for the reader to have a better understand of the achieved difference between these to rates.

b. Add some vertical guidelines in Fig. 3.b of the corresponding frequency of γ_+ and γ_- .

6. In the middle of page 7 the authors claim the drive strength to be " $\Omega/(2\pi) = 8.57$ MHz" and later in Fig. 4 Ω is swept from -10 to 10 MHz. Such drive is usually performed via a microwave source from which the power is set in unit of dBm. Thus, a calibration is required to be able to claim a drive amplitude in MHz, relative to the qubit, instead of dBm, relative to the microwave source. I do think such calibration should be:

a. Mentioned to have be done in the main text.

b. Shown in the supplementary.

7. In the middle of page 7 the authors claim "we are able to prepare an arbitrary qubit state in the XZ-plane of the Bloch sphere". In my opinion, this claim is not currently being supported by their experiment. Indeed in Fig. 3(c) they prove to have reach a single point in the XZ-plane of the Bloch sphere. While I do agree that nothing seem to prevent to reach other points in the plane, I would encourage the author the present, in addition to their current random position shown in Fig. 3c, either:

a. A 2d plot where they present the purity of the reached steady state as function of the (x, z) coordinate. These data will support their claim of "arbitrary" state in a much stronger way than just one random point on the plane.

b. While less convincing, an experiment where they reach a position (0, 0, ~1) and (~1, 0, 0) in the (x, y, z) Bloch sphere. These orthogonal coordinates would make it clearer that they can reach any points in the (x, z) plane by combination.

8. In the middle of page 7, the author wrote "in the limit $t_{\text{drive}} \rightarrow \infty$ ". This infinity seems an unphysical constraint. Moreover, later in the manuscript, a t_{drive} of only $1\mu\text{s}$ is enough. This duration is quite close to any other time/rate discussed in the manuscript and tends to make this infinity requirement not really a requirement.

a. Is there another way of expressing this t_{drive} duration constraint by using some system parameters (Ω , γ , ...)?

b. Surely related to the previous question, why only $1\mu\text{s}$ is enough? How this requirement evolves depending on the drive, qubit and transducer parameters?

9. At the end of page 7, the author wrote: "y including the global dephasing rate $\gamma_\varphi = 0.75\mu\text{s}^{-1}$ and global depolarization rate of $\gamma_1 = 2.565\mu\text{s}^{-1}$ in the numerical solutions to Eq. 2, we find quantitative agreement to the tomography data". It is then unclear if " γ_φ " and " γ_1 " are fit parameters to the data or if " γ_φ " and " γ_1 " have been measured in an independent measurement (typical T1 and T2 Ramsey experiment). Please make this point clearer.

10. At the end of page 7, the author wrote: "indicating that this particular superposition state is cooled to an effective temperature $T_{\text{eff}} \approx 900\mu\text{K}$ ". I do fail to understand the concept of effective

temperature for a superposition state. Looking at ref [40], I fail to link what is reported in the paper and what is discussed in the reference. Please make this point clearer.

11. At the top of page 9, the author wrote: “To investigate this steady-state behavior, we apply a coherent drive to the qubit for a preparation time $t_{\text{prep}} = 3 \mu\text{s}$, which is approximately one order of magnitude longer than the measured depolarization time of the qubit in the absence of drive (see Fig. 4(a))”. I have several concern:

a. Earlier in the main text, the author presented “ γ_1 ” as the “global depolarization rate”. Is this sentence supposed to make us understand that: $t_{\text{prep}} \gg 2 \pi / \gamma_1$?

b. The coherent driving time depicted in Fig. 4.a is far longer than t_{prep} , why is that?

c. Is there any difference between the t_{drive} discussed earlier in the paper and t_{prep} ? It seems to me that both names refer to the same idea.

12. At the bottom of page 9 in Fig.4.b, the colorscale goes from -1 to +1 but on the cut display on Fig.4.c the value plateau well below -1, +1. My question is the following: does the data are close enough to -1, +1 so that such colorscale range is required? If not, for instance -0.8, +0.8, I would prefer to show colorscale closer to the data range to not lead the reader to think that σ_x reaches large values.

13. On the same Fig.4, it is my understanding that there is no fitting parameters at that point of the paper. Is it correct? If so, I would add it in the caption to make it clearer for the reader.

14. In the middle of page 15, the author described the assembly of the hybrid system and, if I understand correctly, explained that the antenna of the qubit chip, here to couple the qubit to the cavity, is mimicked on the saw chip to couple the saw cavity to the qubit. The same antenna design is used to couple, in one hand, the qubit and the 3d microwave cavity and, in the other hand, the qubit and the SAW cavity. My question is the following: Is there any direct coupling between the SAW cavity and the 3d microwave cavity? I would guess yes and that the frequency difference of the 3d cavity, 4.788GHz, and the qubit during the experiments, 4.001GHz, is large enough to claim that such a coupling has a negligible influence on their experiment. I would like the author to comment on that remark and add such comment on the manuscript.

15. On the table S1 please indicate which value is measured and which is by designed. Also this table is referred as a figure previously in the manuscript.

16. On page 3 the equation S6 is refereed in the text. This equation does not exist.

Small remarks:

1. Ref 9 is not about correlated error in superconducting qubit system (there is no qubit) I so fail to see the relevance of that reference.

Reviewer #2 (Remarks to the Author):

This manuscript “Phononic bath engineering of a superconducting qubit” experimentally demonstrates that a piezoelectric surface acoustic wave resonator can be exploited as a strongly frequency-selective dissipation bath for a superconducting qubit. Hybrid quantum systems, like the qubit-mechanical system demonstrated here, are an exciting new area research in which many experiments are asking what the combined system can do that the individual components cannot. While similar qubit bath engineering has previously been demonstrated (by two of the authors) with microwave electrical filters, this is the first demonstration of this protocol with a predominantly mechanical environment.

The manuscript is well-written and shows clear quantitative agreement between experimental measurements and Lindblad master equation theory. While the coherence of the qubit and the SAW modes are relatively modest, there is significant novelty in demonstrating and understanding the power of the combined system. That said, there are a few important questions that I would like the authors to address before I can recommend publication.

1. While this work shows that this method can be used to prepare an arbitrary quantum state of a qubit, it is unclear why this method would be preferred (currently or in the future) over traditional methods. It does not seem like it is faster, higher fidelity or more scalable, so for what reason should this methodology be pursued? Is stabilizing a steady-state more useful or preferable than simply using traditional gate operations to initialize that state?

2. One of claims of the paper is using a “dissipation-enabled stabilization protocol” to achieve a state purity of 85%. While yes, this is much greater than a 50% mixed state, that is not a particularly interesting bar to compare to. For example, what is the purity of the initial state or similar target state prepared with simple gates?

3. The authors also state their superposition state is cooled to 900 uK. This statement needs more explanation than a simple reference to a review of modern physics. In fact, explaining how this effective temperature can be lower than the physical temperature would be useful for articulating the counterintuitive nature of dissipation engineering. For example, in AMO physics sideband cooling protocols routinely use a very high frequency (and very pure) optical transition to cool and increase the purity of a low frequency motional mode. In that case it is clear why the effective temperature of the motion can be lowered...because the ~THz transition is very pure even at room temperature. In this case, it is unclear exactly what degree of freedom is at 900 uK or if the effective starting temperature is hotter or colder than 900 uK. For example a 4 GHz qubit at $T=900$ uK would have an unbelievably small thermal impurity of $<10^{-90}$, so I assume this must be referenced to another energy scale, like that of the Rabi drive. As stated above, it is not that I think the authors are wrong, but simply unclear in the current version of the manuscript.

4. Figure 1: Part of the novelty of this work is how the authors physically coupled this hybrid system. While fig 1b tries to show what this is, it is still incredibly difficult to understand what we are looking at. An additional illustration or differently annotated micrograph would help the reader understand the individual size scales of the qubit, the SAW and where the capacitive coupling arises from in the flip-chip geometry.

Minor comments for clarity and correctness:

A. Page 3: the phrase “telecom frequencies” is ambiguous and useful as that can refer to frequencies from kHz to THz. I would omit or modify.

B. Fig 1c. The electrical circuit is incomplete because there is no connection from the bottom of the qubit to the bottom of the saw chip. While in the real device this is a capacitive connection, the equivalent circuit should either show two capacitors of $C_c/2$ on the top and the bottom, or only C_c on the top and a direct connection between the bottom nodes of the qubit and the SAW.

C. FIG S1b: It is unclear from the text why the conductance of the SAW has the broad frequency dependence shown in fig S1b. Is this a low Q resonant structure? If so, why? Which physical parameters set the center frequency and the width of the conductance.

Reviewer #3 (Remarks to the Author):

The manuscript presents an experimental study of preparing and stabilizing arbitrary quantum states in a driven Transmon qubit by engineering its coupling to bath of piezoelectric SAW phonons. The experimental setup consists of a flux tunable Transmon qubit coupled to a SAW Fabry P´erot cavity fabricated on the surface of single crystal lithium niobate using flip chip technology, additionally the Transmon is also coupled to a 3-D cavity for readout and control. This setup allows the authors to engineer the dissipation of the Transmon into the SAW Fabry P´erot cavity as a function of frequency by engineering the coupling and the spectrum of modes in the cavity. This coupled with the ability to manipulate the qubit through a microwave drive allows the authors to initialize the Transmon qubit in any arbitrary state in the ZX plane with purity ~ 0.85 and stabilize it for times longer than the qubit's intrinsic lifetime. The experimental results are validated against Lindblad master equation numerics.

I find the idea of this experiment, well implemented and, to the best of my knowledge, original. This highly tunable platform opens a novel path to experimentally investigating the non-unitary evolution of quantum states and effective non-Hermitian Hamiltonians. The experimental methodology and the quality of data (including the analysis of the data) are at the highest level in the field, so that the presented findings and conclusions are reliable and robust. However, before recommending the publication, the following points should be clarified or improved.

1. In page 7 of the manuscript the authors include the effect of global depolarization and dephasing rates, how were these rates (γ_1 , γ_Φ) determined? Were they experimentally measured or were they a fit parameter in Eq 2 of the manuscript. If they were experimentally determined additional details for those measurements should be added to the supplementary section. If they were fit parameters then the authors should comment on how those fitted rates compare to the designed parameters.

2. The authors show data and numerical simulations for preparing and stabilizing a particular state on the Bloch sphere in Fig 3. The data shows a purity of ~ 0.85 that matches very well with the numerical calculations. However, it would also be beneficial to know how the purity varies as a function of the stabilized state. In other words it would be helpful to know average purity of the stabilized state to understand better the robustness of the protocol and the use of this technique to stabilize arbitrary states on the Bloch sphere.

3. The data in Fig 3 and 4 show stabilization of quantum states for duration of 1 μ s and 3 μ s respectively. Additionally it would also be helpful to know the upper limit of the time for which a quantum state can be stabilized using this technique and what decides this limit in the current architecture?. This data would help compare this technique to other available techniques to stabilize an arbitrary quantum state for example active quantum error correction.

3. Finally, the authors also mention that by improving the qubit intrinsic quality factor and the transducer design, quantum states can be stabilized with a 99% purity. It would also be helpful if the authors can elaborate more on the second point and mention which part of the current transducer design is limiting the purity to about 85% and how they would go about improving it so that the purity can reach 99%.

We thank the reviewers for their helpful comments and suggested revisions, as well as their interest in our manuscript. We have addressed all of the reviewer comments in our responses below and in the revised manuscript. We provide a point-by-point response to each of the reviewer's comments and a description of the changes made to the manuscript and SI. For clarity we have copied the reviewer comments below (in red) followed by our response (in black). In addition to the revised manuscript we are also uploading a "marked up" version of the pdf with the changes colored in red. We hope that our revised manuscript will be accepted for publication in Nature Communications.

Sincerely,

J.M. Kitzman (on behalf of all authors)

REVIEWER COMMENTS

Reviewer #1 (Remarks to the Author):

The manuscript from J. M. Kitzman, et al. reports on the experimental investigation of a low-level superposition state creation and stabilization via a continuous microwave drive and an engineered dissipative phononic channel. The experiment joins other researches on hybrid quantum system between superconducting qubit, here of the transmon type, and a phononic bath, here a SAW type. It also joins another field of research on creation and stabilization of deterministic quantum state via bath engineering. The main results of the manuscript are shown in Fig.3 and Fig.4 where the authors present the creation and stabilization of a superposition state of their qubit.

According to the abstract, the authors claim:

1. to have achieved "a new type of quantum control over superconducting circuits".
2. to be able to "prepare and stabilize arbitrary qubit states"
3. to have "find that the presence of the energy-dependent loss imparted onto the qubit by the phonons is well described by a master equation treatment of the composite system".

I do not understand the claim 1. The ref. [37], from the same group it seems, already demonstrated the same experiment with engineered photonic loss channel. Is the fact that the loss channel is here phononic deserve such a claim? I would say no since ref. [37] already demonstrated it. This manuscript used the same "type" of quantum control on a different implementation.

The reviewer is correct, the article by *P.M.Harrington et al.* does present bath engineering using a photonic loss channel. The reviewer also correctly states that here the shaped density of states constituting the bath is phononic and can be engineered precisely using the machinery of the coupling of modes theory. We also emphasize that the phononic loss presented in our manuscript is complementary to the photonic loss and our work serves to open a new paradigm for investigating open quantum systems based on superconducting qubits. All that being said, we agree with the reviewers specific conclusion that the novelty is not captured by claim #1. Therefore, we have removed this language and revised the manuscript to better describe the novelty and importance of our results as well as their context in the landscape of open quantum systems research.

I do think the claim 2 to be unsupported by the presented data and is strongly misleading on the achieved results of the experiment. Even the author themselves wrote later in the manuscript to be able to create "arbitrary qubit state in the XZ-plan" leaving a 3rd dimension of the Bloch sphere unexplored. Even worse, the data presented showed the ability to explore the X axis of the Bloch sphere without any data on the Z axis. This is surprising since a previous paper from the same group, ref. [37] showed very similar results but with the data on the Z axis. I currently think that the authors overclaim their ability to create and stabilize and "arbitrary state" and that such claim should be retracted. However I think the author are right to claim such ability in the XZ plan of the Bloch sphere but they should add data about the Z projection to support that claim.

We thank the reviewer for pointing out this possible source of confusion. The reviewer is correct in stating that in our manuscript we show data demonstrating states that have been prepared in the XZ-plane. In general, however, the plane within the Bloch sphere that can be accessed is controlled by the *phase* of the drive signal. Specifically, in the data we have presented in the paper, we have set the phase of the drive such that the qubit eigenstates ultimately lie in the XZ-plane. The chosen drive amplitude (Rabi frequency) and its detuning relative to the qubit then further constrain the qubit eigenstates to a line (axis) in the XZ-plane. Adjusting the phase of the drive rotates the plane on which the eigenstates lie, and subsequent choice of the drive amplitude and detuning selects a line cut through this new plane further constraining the qubit eigenstates to an axis in this plane. We have added language to better describe that our protocol is not merely limited to preparation of states in the XZ-plane.

That being said, we agree that our claim of arbitrary state preparation is insufficiently nuanced. In particular, we have investigated the versatility of the state preparation protocol in the XZ-plane across the experimentally accessible values of the drive parameters and can be used to create high-purity superposition states across a relatively large range of $\langle\sigma_x\rangle$. However, access to large negative values of $\langle\sigma_z\rangle$ is limited in this device by the difference between γ_+ and γ_- arising from the slope of the SAW device conductance versus frequency. Therefore we have modified the language in the abstract and text regarding claim #2 and have also modified Figure 4 and added additional text to describe more completely the accessibility as well as the limitations of this type of phononic dissipation enabled state preparation.

The claim 3 is surely true for the author but is not new for the community since ref. [21] published in 2018 already saw the conformity between theory and experiments on such hybrid system by using the Lindblad master equation. Such sentence should then be removed.

We thank the reviewer for pointing out the potential confusion regarding this point. We agree that the paper by *K.J. Satzinger et al.* shows agreement between their measurements and a master equation treatment analysis. Additionally, it was not our intention to claim to be the first to apply this type of theory to understanding SAW-qubit hybrid systems. Rather, our intention was a state that we had performed such an analysis to understand our drive-dissipative experimental results and have modified the claim in the abstract to better reflect this fact.

To conclude, the reported experiment is a repetition of a previous experiment of the same group, ref. [37], the difference being the replacement of an engineered photonic loss channel by a photonic one. While this in itself is an achievement and is interesting, the manuscript is strongly spoiled by various unfounded claims. In its current state, this manuscript should not be published in any journal. I do however think that the experiment in itself is of great interest should be published without the current overclaims and lack of data.

In addition to our responses above, we thank the reviewer for their thorough and frank assessment of our work. Additionally, where relevant in our responses below, we further address the reviewer's comments regarding the claims we present in the abstract and main text of the paper. As the reviewer will see from our responses, we often agree with their point of view and have significantly

revised the text to better describe the novelty and importance of our results as well as their context in the landscape of open quantum systems research.

In the following, I give various remarks and question for a possible resubmission:

1. In the abstract, the author claim to be “able to prepare and stabilize arbitrary qubit states”. It is my opinion that this claim is currently not supported by the presented results and therefore should either be retracted or additional data must be presented. Indeed, the main text present the preparation, not stabilization, of arbitrary states in the XZ plane only, see Fig. 3c. Moreover even such a claim is unsupported by their data, see another of my comment later, point 7.

As we have addressed above, the reviewer is correct in stating that in our manuscript we show data demonstrating states that have been prepared in the XZ-plane. In general, however, the plane within the Bloch sphere that can be accessed is controlled by the *phase* of the drive signal. We have added language to better describe that our experiments are not merely limited to preparation of states in the XZ-plane.

Also, as we've addressed above, we generally agree with the reviewer that our claim of arbitrary state preparation is insufficiently nuanced. In particular, we have investigated the versatility of the state preparation protocol in the XZ-plane across the experimentally accessible values of the drive parameters and can be used to create high-purity superposition states across a relatively large range of $\langle\sigma_x\rangle$. However, access to large negative values of $\langle\sigma_z\rangle$ is limited in this device by the difference between γ_+ and γ_- arising from the slope of the SAW device conductance versus frequency. Therefore we have modified the language in the abstract and text regarding claim #2 and have also modified Figure 4 and added additional text to describe more completely the accessibility as well as the limitations of this type of phononic dissipation enabled state preparation.

2. On Fig 1.c the author depicted the SAW device, in blue, to have one capacitive coupling, in black, with the qubit, in salmon. When I look at the device picture shown on Fig. 1.b I see two capacitive coupling pads and so I would have expected the same in Fig. 1.c. Can the author comment on this?

We thank the reviewer for noticing this and they are correct in their understanding that the capacitive coupling is physically distributed between the

two sets of antenna pads. We have modified Fig 1c (the effective circuit), to reflect this fact.

3. On the top paragraph of page 5, the authors claim “The qubit and the confined SAW mode interact with a coupling rate $gm/(2\pi) = 12 \pm 0.6$ MHz, larger than the loss rate of either system, which is a hallmark of the quantum acoustic strong coupling regime.”. There is however, at that point of the manuscript, no demonstration of the qubit lifetime nor the SAW cavity lifetime and while I agree that an avoided level crossing is a visual proof of the strong coupling regime a quantitative argument is still required to make such a claim.

It is possible to find the qubit decay rate later on at the end of page 7. There, the author discussed the “phononic bath engineering protocol” and cite the “global depolarization rate” to be about $2.565 \mu\text{s}^{-1}$, which I assume to be equivalent to a decay rate $2.565/(2\pi) = 0.41$ MHz.

I was unable to find the cavity decay rate in neither the main text nor the supplementary information. It may be possible to deduce the cavity loss rate from the supplementary material where the “SAW propagation loss” η and “speed of sound” v_{sound} is given by assuming an exponential decay behavior (expected for a SAW cavity mode) $\gamma_{\text{saw}}/(2\pi) = 1/(\eta v_{\text{sound}}) = 1.82$ MHz.

Thus while I was, maybe, able to back-up the claim of the author I think that:

- a. The qubit and cavity decay rate should be made clearer to allow an easy comparison with the coupling rate and consequently ensure the achievement of the strong coupling regime.
- b. Such rate being measured, I would add such measurements in the supplementary.

We agree and have added information to this portion of the manuscript regarding the loss rates of the qubit and SAW modes to show that the coupling is in fact larger than either of these loss rates.

Specifically, the qubit decay rate at 4.46GHz is $\gamma_q/(2\pi) = f_q/Q_i = 4.46\text{GHz}/1.67 \times 10^3 = 2.67$ MHz, where f_q is the qubit frequency and Q_i is the bare qubit quality factor. We have modified the manuscript so that the decay rate of the bare qubit is mentioned immediately after the discussion regarding the coupling strength so that this is clear to the reader.

In order to quantify the decay rate of the SAW mode, we fit the response of the simulated SAW resonator near resonance to a Lorentzian function and obtain a decay rate of $\gamma_{\text{SAW}}/(2\pi) = 0.3$ MHz from the full-width at half-maximum. We have also added this information to the main text of the manuscript.

4. On the top paragraph of page 5, the author claimed to have reached the strong coupling regime between the qubit and the SAW cavity. However, I fail to see the relevance of such a result in this manuscript. Indeed the demonstration of the strong coupling regime is not the main message of the paper and has already been demonstrated in ref [21] on a very similar system. Moreover the bath engineering described by the author does not seem to require a cavity, which is here only to offer a large decay rate gradient in a narrow frequency range outside of its bandwidth,

As the reviewer points out, the strong coupling regime is not a requirement for the driven-dissipative experiments we report. Rather we intend it to serve as a verification of the functionality (i.e. hybridization of the qubit and SAW modes) of our device. To better reflect this point we have added language referencing the previous demonstrations of the quantum acoustic strong coupling regime to this portion of the revised manuscript. Additionally, the spectroscopy shown in Fig. 2(b,c) reveal signatures of controlled surface phonon loss arising from the interaction of the qubit with the continuum of SAW modes on either side of the confined SAW mode, which serve to highlight the relevance of phonon-induced loss in our device.

5. In the middle of page 7, the author explained that they chose $\omega_q/(2\pi) = 4.001$ GHz. I do think that such frequency was chosen so that γ_{-1} and γ_{-} is minimum while γ_{+} is maximum. I would

- Add the value of γ_{+} and γ_{-} for the reader to have a better understand of the achieved difference between these to rates.
- Add some vertical guidelines in Fig. 3.b of the corresponding frequency of γ_{+} and γ_{-} .

We agree that this information would be helpful to the reader to understand the experiment more clearly. We have modified Fig. 3b to indicate the endpoints of the experimentally accessible sidebands of the Mollow triplet and have included the relevant decay rates in the manuscript.

6. In the middle of page 7 the authors claim the drive strength to be " $\Omega/(2\pi) = 8.57$ MHz" and later in Fig. 4 Ω is swept from -10 to 10 MHz. Such drive is usually performed via a microwave source from which the power is set in unit of

dBm. Thus, a calibration is required to be able to claim a drive amplitude in MHz, relative to the qubit, instead of dBm, relative to the microwave source. I do think such calibration should be:

- a. Mentioned to have been done in the main text.
- b. Shown in the supplementary.

We agree and have made both of these changes. Specifically we have mentioned the calibration in the manuscript and added it to the supplemental information.

7. In the middle of page 7 the authors claim “we are able to prepare an arbitrary qubit state in the XZ-plane of the Bloch sphere”. In my opinion, this claim is not currently being supported by their experiment. Indeed in Fig. 3(c) they prove to have reached a single point in the XZ-plane of the Bloch sphere. While I do agree that nothing seems to prevent reaching other points in the plane, I would encourage the author to present, in addition to their current random position shown in Fig. 3c, either:

- a. A 2d plot where they present the purity of the reached steady state as a function of the (x, z) coordinate. These data will support their claim of “arbitrary” state in a much stronger way than just one random point on the plane.
- b. While less convincing, an experiment where they reach a position $(0, 0, \sim 1)$ and $(\sim 1, 0, 0)$ in the (x, y, z) Bloch sphere. These orthogonal coordinates would make it clearer that they can reach any points in the (x, z) plane by combination.

As we have already addressed above, the reviewer is correct in stating that in our manuscript we show data demonstrating states that have been prepared in the XZ-plane. In general, however, the plane within the Bloch sphere that can be accessed is controlled by the *phase* of the drive signal. We have added language to better describe that our experiments are not merely limited to preparation of states in the XZ-plane.

Again we generally agree with the reviewer that our claim of arbitrary state preparation is insufficiently nuanced. In particular, we have investigated the versatility of the state preparation protocol in the XZ-plane across the experimentally accessible values of the drive parameters and can be used to create high-purity superposition states across a relatively large range of $\langle \sigma_x \rangle$. However, access to large negative values of $\langle \sigma_z \rangle$ is limited in this device by the difference between γ_+ and γ_- arising from the slope of the SAW device conductance versus frequency. Therefore we have modified the language in the abstract and text regarding claim #2 and have also modified Figure 4 and added

additional text to describe more completely the accessibility as well as the limitations of this type of phononic dissipation enabled state preparation.

8. In the middle of page 7, the author wrote “in the limit $t_{\text{drive}} \rightarrow \infty$ ”. This infinity seems an unphysical constraint. Moreover, later in the manuscript, a t_{drive} of only $1\mu\text{s}$ is enough. This duration is quite close to any other time/rate discussed in the manuscript and tends to make this infinity requirement not really a requirement.

a. Is there another way of expressing this t_{drive} duration constraint by using some system parameters (Ω, γ, \dots)?

b. Surely related to the previous question, why only $1\mu\text{s}$ is enough? How this requirement evolves depending on the drive, qubit and transducer parameters?

We agree with the reviewer that this timescale should be explained more thoroughly. In particular, for the protocol to work the drive duration needs to be longer than T_1 of the qubit, which is dominated by decay into phonons in our device. In the data presented in Fig. 3c,d the qubit lifetime is $T_1 = 185\text{ ns}$. By performing Rabi oscillations over a duration of $1\mu\text{s}$, we guarantee that in the absence of the dynamical state stabilization protocol, any excitation would have caused the qubit to relax into its ground state. We have modified the language in the manuscript to emphasize that “ $t_{\text{drive}} \gg T_1$ ”.

9. At the end of page 7, the author wrote: “y including the global dephasing rate $\gamma_{\phi} = 0.75\mu\text{s}^{-1}$ and global depolarization rate of $\gamma_{-1} = 2.565\mu\text{s}^{-1}$ in the numerical solutions to Eq. 2, we find quantitative agreement to the tomography data”. It is then unclear if “ γ_{ϕ} ” and “ γ_{-1} ” are fit parameters to the data or if “ γ_{ϕ} ” and “ γ_{-1} ” have been measured in an independent measurement (typical T_1 and T_2 Ramsey experiment). Please make this point clearer.

These rates are fit parameters to the data. We are also able to estimate these parameters from experimental data and find reasonable agreement between the two methods. We have added language to the revised manuscript to clarify this point. Additionally we have included the measurements for T_1 and T_2 Ramsey experiments to the supplementary information.

10. At the end of page 7, the author wrote: “indicating that this particular superposition state is cooled to an effective temperature $T_{\text{eff}} = 900\mu\text{K}$ ”. I do fail to understand the concept of effective temperature for a superposition state. Looking at ref [40], I fail to link what is reported in the paper and what is discussed in the reference. Please make this point clearer.

We agree and have extensively expanded this discussion about the effective temperature in the revised manuscript. In particular, we have introduced a simple model that does not require Ref. [40], which we have removed from the revised manuscript. To summarize, we model the qubit as a two level system in thermal equilibrium with an effective phononic bath. By calculating the mean energy of the qubit in the dressed basis from tomography measurements and comparing to the mean energy given from the partition function, we subsequently calculate an effective temperature of $T_{\text{eff}} \approx 250 \mu\text{K}$. As noted above, we have added a more complete discussion of this model to the revised manuscript. We also note that during the process of expanding this model, and checking our calculations, we discovered an error in our previous calculation, hence the difference in the effective temperature (i.e. $250 \mu\text{K}$ versus $900 \mu\text{K}$).

11. At the top of page 9, the author wrote: "To investigate this steady-state behavior, we apply a coherent drive to the qubit for a preparation time $t_{\text{prep}} = 3 \mu\text{s}$, which is approximately one order of magnitude longer than the measured depolarization time of the qubit in the absence of drive (see Fig. 4(a))". I have several concerns:

- a. Earlier in the main text, the author presented " γ_1 " as the "global depolarization rate". Is this sentence supposed to make us understand that: $t_{\text{prep}} \gg 2\pi/\gamma_1$?
 - b. The coherent driving time depicted in Fig. 4.a is far longer than t_{prep} , why is that?
 - c. Is there any difference between the t_{drive} discussed earlier in the paper and t_{prep} ? It seems to me that both names refer to the same idea.
- a. To be clear, γ_1 refers to the qubit decay into non-SAW phonon sources, while Γ_q includes (and is dominated by) SAW phonons. That being said, the reviewer's intuition is correct here, the drive time should be much longer than the qubit depolarization time. We have modified the language in the manuscript to emphasize that " $t_{\text{drive}} \gg T_1$ ".
 - b. The length of the arrow indicating the coherent driving time depicted in Fig. 4a of the original manuscript was simply wrong. We've fixed this error in the updated manuscript.
 - c. The reviewer is correct, these are the same parameter. In the updated manuscript we have exclusively refer to only t_{drive} .

12. At the bottom of page 9 in Fig.4.b, the colorscale goes from -1 to +1 but on the cut display on Fig.4.c the value plateau well below -1, +1. My question is the following: does the data are close enough to -1, +1 so that such colorscale range is required? If not, for instance -0.8, +0.8, I would prefer to show colorscale closer to the data range to not lead the reader to think that σ_x reaches large values.

The data does not extend fully from -1 to +1. We have modified the colorbar to span from -0.8 to +0.8 on all 2D plots containing tomography data as the reviewer suggests.

13. On the same Fig.4, it is my understanding that there is no fitting parameters at that point of the paper. Is it correct? If so, I would add it in the caption to make it clearer for the reader.

As discussed above, the rates γ_1 and γ_ϕ are determined from fits to the data. Otherwise all the other parameters in the model are determined empirically. We have added a sentence to this effect in the caption to the updated Fig. 4.

14. In the middle of page 15, the author described the assembly of the hybrid system and, if I understand correctly, explained that the antenna of the qubit chip, here to couple the qubit to the cavity, is mimicked on the saw chip to couple the saw cavity to the qubit. The same antenna design is used to couple, in one hand, the qubit and the 3d microwave cavity and, in the other hand, the qubit and the SAW cavity. My question is the following: Is there any direct coupling between the SAW cavity and the 3d microwave cavity? I would guess yes and that the frequency difference of the 3d cavity, 4.788GHz, and the qubit during the experiments, 4.001GHz, is large enough to claim that such a coupling has a negligible influence on their experiment. I would like the author to comment on that remark and add such comment on the manuscript.

The reviewer is correct that there is direct coupling between the 3D cavity and the SAW device. In fact, in separate experiments, we are able to directly populate the SAW device with phonons by applying a drive resonant with the main SAW mode. However, in the measurements presented in the manuscript, all applied signals are far detuned from the main resonance of the confined SAW mode and we therefore expect this mode to remain near its ground state. We have added a sentence to the revised manuscript to this effect.

15. On the table S1 please indicate which value is measured and which is by designed. Also this table is referred as a figure previously in the manuscript.

We have indicated in the revised SI the values are determined from the data versus those that are fabrication design parameters and have clarified the references to Table S1 and Figure S1 in the SI.

16. On page 3 the equation S6 is refereed in the text. This equation does not exist.

We thank the reviewer for catching this typo. This should have read Eq. S4, and we have corrected this in the updated SI.

Small remarks:

1. Ref 9 is not about correlated error in superconducting qubit system (there is no qubit) I so fail to see the relevance of that reference.

We have removed this reference in the updated manuscript.

Reviewer #2 (Remarks to the Author):

This manuscript “Phononic bath engineering of a superconducting qubit” experimentally demonstrates that a piezoelectric surface acoustic wave resonator can be exploited as a strongly frequency-selective dissipation bath for a superconducting qubit. Hybrid quantum systems, like the qubit-mechanical system demonstrated here, are an exciting new area research in which many experiments are asking what the combined system can do that the individual components cannot. While similar qubit bath engineering has previously been demonstrated (by two of the authors) with microwave electrical filters, this is the first demonstration of this protocol with a predominantly mechanical environment.

The manuscript is well-written and shows clear quantitative agreement between experimental measurements and Lindblad master equation theory. While the coherence of the qubit and the SAW modes are relatively modest, there is significant novelty in demonstrating and understanding the power of the combined system. That said, there are a few important questions that I would like the authors to address before I can recommend publication.

We thank the reviewer for their thorough review of our manuscript and interest in our work. Below we provide a point by point response to each of the questions/comments posed by the reviewer.

1. While this work shows that this method can be used to prepare an arbitrary quantum state of a qubit, it is unclear why this method would be preferred (currently or in the future) over traditional methods. It does not seem like it is faster, higher fidelity or more scalable, so for what reason should this methodology be pursued? Is stabilizing a steady-state more useful or preferable than simply using traditional gate operations to initialize that state?

We thank the reviewer for this insightful question. As the reviewer correctly states, there exist a wide variety of high-fidelity quantum state preparation protocols, some of which demonstrate routes toward scalable implementation. On the other hand, the novelty/importance of our work is in the fundamental investigation and understanding of the physics of driven-dissipative quantum acoustics systems. Therefore, we have revised the manuscript to better describe the novelty and importance of our results as well as their context in the landscape of open quantum systems research. The results advance the understanding of mechanical losses in open quantum systems and extensions

of this work could impact the mitigation of phonon-induced decoherence in superconducting processors.

2. One of claims of the paper is using a “dissipation-enabled stabilization protocol” to achieve a state purity of 85%. While yes, this is much greater than a 50% mixed state, that is not a particularly interesting bar to compare to. For example, what is the purity of the initial state or similar target state prepared with simple gates?

The reviewer is correct that it is possible to prepare higher purity states using traditional gate operations. That being said, the ability to use phononic loss in a driven system to dynamically stabilize quantum states offers a unique pathway to state preparation, the fidelity of which should be investigated. We have added language to describe, in more detail, how improvements to the qubit and SAW device quality can be used to improve the efficiency of the dissipation-enabled dynamical stabilization. Additionally, as discussed in the response to the previous point, our results offer significant insight into the fundamental role of phononic dissipation in superconducting qubit systems. In the revised manuscript we have attempted to better articulate this point of view.

3. The authors also state their superposition state is cooled to 900 uK. This statement needs more explanation than a simple reference to a review of modern physics. In fact, explaining how this effective temperature can be lower than the physical temperature would be useful for articulating the counterintuitive nature of dissipation engineering. For example, in AMO physics sideband cooling protocols routinely use a very high frequency (and very pure) optical transition to cool and increase the purity of a low frequency motional mode. In that case it is clear why the effective temperature of the motion can be lowered...because the ~THz transition is very pure even at room temperature. In this case, it is unclear exactly what degree of freedom is at 900 uK or if the effective starting temperature is hotter or colder than 900 uK. For example a 4 GHz qubit at $T=900$ uK would have an unbelievably small thermal impurity of $<10^{-90}$, so I assume this must be referenced to another energy scale, like that of the Rabi drive. As stated above, it is not that I think the authors are wrong, but simply unclear in the current version of the manuscript.

We thank the reviewer for bringing up this very interesting aspect of our work. The reviewer is correct that the effective temperature must be referenced to the energy scale of the Rabi drive and is substantially lower than the base temperature of the dilution refrigerator. We have extensively expanded the

discussion regarding the effective temperature in the revised manuscript. In particular, we have introduced a simple model of the qubit as a two level system in thermal equilibrium with an effective phononic bath. By calculating the mean energy of the qubit in the dressed basis from tomography measurements and comparing to the mean energy given from the partition function, we subsequently calculate an effective temperature of $T_{\text{eff}} \approx 250 \mu\text{K}$. As noted above, we have added a more complete discussion of this model to the revised manuscript. We also note that during the process of expanding this model, and checking our calculations, we discovered an error in our previous calculation, hence the difference in the effective temperature (i.e. $250 \mu\text{K}$ versus $900 \mu\text{K}$).

4. Figure 1: Part of the novelty of this work is how the authors physically coupled this hybrid system. While fig 1b tries to show what this is, it is still incredibly difficult to understand what we are looking at. An additional illustration or differently annotated micrograph would help the reader understand the individual size scales of the qubit, the SAW and where the capacitive coupling arises from in the flip-chip geometry.

We have made several modifications to the revised manuscript and SI to address this point. 1) We have updated Fig. 1 in the main manuscript to better show how the coupling capacitances are physically distributed in the device and also included a larger image of the SAW resonator. 2) We have included a schematic in the supplemental information which expands upon the relevant length scales of the hybrid system and details the coupling mechanism.

Minor comments for clarity and correctness:

A. Page 3: the phrase “telecom frequencies” is ambiguous and useful as that can refer to frequencies from kHz to THz. I would omit or modify.

We have removed the phrase “at telecom frequencies” from the manuscript.

B. Fig 1c. The electrical circuit is incomplete because there is no connection from the bottom of the qubit to the bottom of the saw chip. While in the real device this is a capacitive connection, the equivalent circuit should either show two capacitors of $C_c/2$ on the top and the bottom, or only C_c on the top and a direct connection between the bottom nodes of the qubit and the SAW.

We thank the reviewer for noticing this and they are correct that the capacitive coupling is physically distributed between the two sets of antenna pads. We have modified Fig 1c (the effective circuit), to reflect this fact.

C. FIG S1b: It is unclear from the text why the conductance of the SAW has the broad frequency dependence shown in fig S1b. Is this a low Q resonant structure? If so, why? Which physical parameters set the center frequency and the width of the conductance.

The broad frequency response of the conductance in Fig. S.1c arises from the Fourier transform of the spatial structure of the SAW transducer (i.e. the central IDT). In particular, the width of the response is dictated by the number of finger pairs constituting the transducer, labeled N_p in Table S1. We have added a description of this to the SI.

Reviewer #3 (Remarks to the Author):

The manuscript presents an experimental study of preparing and stabilizing arbitrary quantum states in a driven Transmon qubit by engineering its coupling to bath of piezoelectric SAW phonons. The experimental setup consists of a flux tunable Transmon qubit coupled to a SAW Fabry P´erot cavity fabricated on the surface of single crystal lithium niobate using flip chip technology, additionally the Transmon is also coupled to a 3-D cavity for readout and control. This setup allows the authors to engineer the dissipation of the Transmon into the SAW Fabry P´erot cavity as a function of frequency by engineering the coupling and the spectrum of modes in the cavity. This coupled with the ability to manipulate the qubit through a microwave drive allows the authors to initialize the Transmon qubit in any arbitrary state in the ZX plane with purity ~ 0.85 and stabilize it for times longer than the qubit's intrinsic lifetime. The experimental results are validated against Linblad master equation numerics.

I find the idea of this experiment, well implemented and, to the best of my knowledge, original. This highly tunable platform opens a novel path to experimentally investigating the non-unitary evolution of quantum states and effective non-Hermitian Hamiltonians. The experimental methodology and the quality of data (including the analysis of the data) are at the highest level in the field, so that the presented findings and conclusions are reliable and robust. However, before recommending the publication, the following points should be clarified or improved.

We thank the reviewer for their concise summary and positive interest in our work. Below we provide a point by point response to all comments posed by the reviewer.

1. In page 7 of the manuscript the authors include the effect of global depolarization and dephasing rates, how were these rates (γ_1, γ_ϕ) determined? Were they experimentally measured or were they a fit parameter in Eq 2 of the manuscript. If they were experimentally determined additional details for those measurements should be added to the supplementary section. If they were fit parameters then the authors should comment on how those fitted rates compare to the designed parameters.

These two rates are fit parameters to the data. Additionally, we are also able to determine these parameters from the experimental data and find reasonable agreement between the two methods. We have added language to the revised

manuscript to clarify this point and have also included the measurements for T_1 and T_2 Ramsey experiments to the supplementary information, which allow us to determine these parameters from the experiment.

2. The authors show data and numerical simulations for preparing and stabilizing a particular state on the Bloch sphere in Fig 3. The data shows a purity of ~ 0.85 that matches very well with the numerical calculations. However, it would also be beneficial to know how the purity varies as a function of the stabilized state. In other words it would be helpful to know average purity of the stabilized state to understand better the robustness of the protocol and the use of this technique to stabilize arbitrary states on the Bloch sphere.

We have modified Fig. 4 to include tomography data showing the purity along both the z and x axes of the Bloch sphere (color scale indicating the state purity). Additionally we have expanded the discussion about the data in Fig. 4 to describe the accessible regimes, as well as limitations, to the dynamically stabilized states. We have also included the tomography data along the z axis and the state purity as a function of drive parameters in the supplemental information.

3. The data in Fig 3 and 4 show stabilization of quantum states for duration of 1 μ s and 3 μ s respectively. Additionally it would also be helpful to know the upper limit of the time for which a quantum state can be stabilized using this technique and what decides this limit in the current architecture?. This data would help compare this technique to other available techniques to stabilize an arbitrary quantum state for example active quantum error correction.

We emphasize that in the driven basis, the qubit relaxes to its effective ground state for the duration of the drive. Immediately after cessation of the drive the qubit state is measured in the undriven basis and the superposition state of the qubit is recovered. However, it should be noted, that once the drive is turned off the natural decay mechanisms of the qubit (i.e. those present in the lab frame) dominate the state evolution and the state information is lost. We have modified the revised manuscript to better emphasize that the resulting states that are stabilized are done so in a dynamical fashion under the combined presence of drive and SAW phonon dissipation.

3. Finally, the authors also mention that by improving the qubit intrinsic quality factor and the transducer design, quantum states can be stabilized with a 99% purity. It would also be helpful if the authors can elaborate more on the second

point and mention which part of the current transducer design is limiting the purity to about 85% and how they would go about improving it so that the purity can reach 99%.

We have expanded the discussion in the conclusion of the manuscript to describe in more detail the kinds of improvements that can be made to enhance the dynamically stabilized state purity. For completeness, the primary factors that dictate the final purity of the stabilized state are 1) the gradient of the qubit loss as a function of frequency. 2) By decreasing the global decay rates Υ_1 and Υ_ϕ , the expected state purity increases, which is achievable with a higher quality qubit.

REVIEWER COMMENTS

Reviewer #1 (Remarks to the Author):

I thank the authors for their thorough answers to my comments. I do find that the quality of the paper has greatly improved in the sense that much of its over-claims have been removed (one stay, I think, see my comments afterwards) and exposed facts are better explained. I thank the authors to have added data in the supplementary. I do find that it adds support to their paper. The added data are clear and well explained. The Fig.S4.a,b is a nice add-on.

As In my previous review I do find that experiment to be of great interest and should be published but not it its current state, mainly for an unsupported claim about the purity at the end of the paper (see my comments).

In the following, I give various remarks and question for a possible resubmission:

1. Between the two versions of the paper many rates have been modified. For instance before $\gamma\phi = 0.75 \mu\text{s}^{-1}$ (page 7) and now $\gamma\phi = 1.48 \mu\text{s}^{-1}$ (page 8).

a. Why these changes?

2. At the top of page 5, the authors wrote:

“Near this confined acoustic mode, we fit the simulated conductance to a Lorentzian function and extract the SAW decay rate $\gamma_{\text{SAW}}/(2\pi) = 0.3 \text{ MHz}$ as the full-width at half-maximum”.

a. It is unclear if the author took γ as the full-width at half-maximum, energy decay, or as the half-width at half-maximum, amplitude decay. Please make it clearer.

b. I would add this fit to the supplementary.

3. On page 8, the authors explained how they obtained the effective temperature of 250uK for their qubit. Their simple model states that $T_{\text{eff}} = -\hbar \Omega_r / (2 k_b \text{arctanh}(\langle \sigma_z \rangle))$. I derived myself the formula using textbook quantum statistics reference and agree with the obtained formula. I also got the same effective temperature of 250uK using the data of Fig.3.d.

a. I would encourage the authors to precise if this measurement correctly showed the coolest obtainable temperature of their setup.

b. In addition, what is currently limiting their cooling ability?

c. From the data of Fig.3.d, I got $\langle \sigma_x \rangle \sim -0.58$ and $\langle \sigma_y \rangle \sim 0.56$ which gives $\langle \tilde{\sigma}_z \rangle \sim -0.80$ equivalent of a qubit thermal population of $\sim 10\%$ ($\Omega_r / (2 \pi) = 13 \text{ MHz}$). If these results are correct, I would like the authors to add some of these information on the paper so that it is clear for the readers that the dynamically stabilized qubit:

- i. is of “low” frequency compare to the usual one, for instance a transmon is usually around 5GHz.
- ii. calculated “low” temperature is comparable to a transmon temperature of $\sim 110\text{mK}$ if we take a 5GHz transmon thermally populated a 10%,.

My point here is that I think adding a context to the obtained effective temperature would be beneficial to the paper. Of course feel free to use another angle if you wish, my comparison with a transmon was done to illustrate my point only and not to impose it.

4. On page 10, the author wrote: “We tomographically reconstruct the steady-state expectation value $\langle \sigma_{x,z} \rangle = \text{Tr}(\rho \sigma_{x,z})$ in the bare qubit eigenbasis as a function of drive parameters and compare with the solutions to Eq. 2 as shown in Fig. 4(a,b)”.

a. The Fig.4.a and Fig.4.b only show σ_x . Please correct the text.

5. On page 11, the author wrote: “As we modify the coupling between the qubit and SAW-phonon bath, we observe excellent agreement between the measured and predicted purity of the resulting qubit state”.

a. I do not understand how the author can discuss the purity of the obtained state with a figure showing the expected value $\langle \sigma_x \rangle$. Maybe this sentence referred to the sup Fig.S4 which would make sense.

b. Please explain.

6. On page 11, the author wrote: “We find that the results shown in Fig.4.c are also in good agreement with the open quantum systems modeling of the hybrid systems based on Eq. 2”.

a. This assertion is currently not supported the paper.

b. I agree that with the model shown in Fig.3.e as a solid red line (solid red line not explained to the reader) it seems that purity reached for long drive duration is understood. However, this figure only prove it for one set of drive parameters, not for the whole sets shown in Fig.4.c. I would encourage the author to add proof of their claim.

Typo:

1. In the sup page 4, I think the author mean Eq. 2 and not 4.

Reviewer #3 (Remarks to the Author):

All points raised by me during the initial review have been adequately addressed. That being said, I would recommend this work for publication.

We thank the reviewers for their additional helpful comments and suggested revisions, as well as their continued interest in our manuscript. In this second round of review we have addressed all of the reviewer comments in our responses below and in the revised paper. We provide a point-by-point response to each of the reviewer's comments and a description of the changes made to the manuscript and SI. For clarity we have copied the reviewer comments below (in red) followed by our response (in black). In addition to the revised manuscript we are also uploading a "marked up" version of the pdf with the changes colored in red. We hope that our newly revised manuscript will be accepted for publication in Nature Communications.

Sincerely,

J.M. Kitzman + J. Pollanen (on behalf of all authors)

REVIEWER COMMENTS

Reviewer #1 (Remarks to the Author):

I thank the authors for their thorough answers to my comments. I do find that the quality of the paper has greatly improved in the sense that much of its over-claims have been removed (one stay, I think, see my comments afterwards) and exposed facts are better explained. I thank the authors to have added data in the supplementary. I do find that it adds support to their paper. The added data are clear and well explained. The Fig.S4.a,b is a nice add-on.

As In my previous review I do find that experiment to be of great interest and should be published but not it its current state, mainly for an unsupported claim about the purity at the end of the paper (see my comments).

We thank the reviewer for their second review of the manuscript and continued interest in our work. In the revised version we have expanded the discussion of the results to better support our claims regarding the state purity and modified the figures to more clearly present these results and compare them to solutions to the Lindblad master equation (see detailed responses below). We hope these changes address the reviewer's primary concern regarding this point. Additionally, we have addressed all of the reviewers' other technical points and typos (see detailed responses below).

In the following, I give various remarks and question for a possible resubmission:

1. Between the two versions of the paper many rates have been modified. For instance before $\gamma\phi = 0.75 \mu\text{s}^{-1}$ (page 7) and now $\gamma\phi = 1.48 \mu\text{s}^{-1}$ (page 8).

a. Why these changes?

We thank the reviewer for taking the time to double check these rates.

Between the two versions the value of $\gamma\phi$ obtained from fitting the data changed because we modified Eq. 2 in the main text so that the prefactor in front of the last term was changed from $\gamma\phi$ to $\gamma\phi/2$. This was done to conform to a convention that is more typical when presenting the master equation (see e.g. *Satzinger et al. Nature 2018* and *Harrington et al. PRA 2019*, which are Refs. [20, 36] in the current version of the manuscript). Additionally we have added the subscript “exp” to the experimentally determined value of “ $\gamma\phi_{\text{exp}}$ ” to distinguish it from the fitted rate $\gamma\phi$.

The rate γ_1 changed slightly (by 4%) from version 1 to version 2 because we optimized the fitting routine used to fit the data to Eq. 2 when we reanalyzed the data in responding to comment #9 by Reviewer 1 and comment #1 by Reviewer 3 in the first round of review.

The rate $\Omega/(2\pi)$ on the top of page 8 of the current version of the manuscript was changed because of a typo in version 1 of the manuscript that was corrected in version 2. The correct value is 8.47 MHz.

Finally, the internal qubit quality factor listed on the bottom of page 6 of the current version of the manuscript changed because we identified and added a missing factor of 2π in Eq. 1 after the first round of reviews.

No other rates changed between the first two versions of the manuscript.

2. At the top of page 5, the authors wrote:

“Near this confined acoustic mode, we fit the simulated conductance to a Lorentzian function and extract the SAW decay rate $\gamma_{\text{SAW}}/(2\pi) = 0.3 \text{ MHz}$ as the full-width at half-maximum”.

a. It is unclear if the author took γ as the full-width at half-maximum, energy decay, or as the half-width at half-maximum, amplitude decay. Please make it clearer.

We thank the Reviewer for catching this mistake. We originally intended to use the half-width at half maximum of 0.3 MHz, but accidentally referred to it as the full-width at half maximum. However, upon reconsideration we think the relevant quantity is the energy decay rate of the phonon mode, which is the full width at half maximum. We have modified the language in the main text and supplemental information of the paper to make this point clear, i.e. that we are using the full width at half maximum = 0.6 MHz that corresponds to the energy decay rate of the phonon mode.

b. I would add this fit to the supplementary.

We have added this fit to Supplemental Figure S1(d).

3. On page 8, the authors explained how they obtained the effective temperature of 250uK for their qubit. Their simple model states that $T_{\text{eff}} = -\hbar \Omega_r / (2 k_b \text{arctanh}(\langle \sigma_z \rangle))$. I derived myself the formula using textbook quantum statistics reference and agree with the obtained formula. I also got the same effective temperature of 250uK using the data of Fig.3.d.

a. I would encourage the authors to precise if this measurement correctly showed the coolest obtainable temperature of their setup.

Yes, we find that the temperature of 250 uK is the lowest effective temperature that we can prepare using the driven-dissipative protocol in our setup. We have added a sentence to the main text that states this.

b. In addition, what is currently limiting their cooling ability?

This limitation is likely due to the fact that the qubit used in our device has a relatively low intrinsic coherence (i.e. low energy relaxation time and phase coherence time). To try and understand this limitation we can use the simulation based on Eq. 2, which indicates that decreasing the rates γ_ϕ and γ_1 leads to lower effective temperatures and thus a greater cooling ability. In particular, by setting $\gamma_\phi = \gamma_1 = 0.1 \mu\text{s}^{-1}$ (i.e. if the intrinsic qubit energy relaxation and dephasing were both better by approximately an order of magnitude), the solutions to the master equation predict that we could reach

an effective temperature as low as 85 uK with all the other parameters the same as our existing experiment. We have added language that describes this to the manuscript.

c. From the data of Fig.3.d, I got $\langle \sigma_x \rangle \sim -0.58$ and $\langle \sigma_y \rangle \sim 0.56$ which gives $\langle \tilde{\sigma}_z \rangle \sim -0.80$ equivalent of a qubit thermal population of $\sim 10\%$ ($\Omega_r / (2\pi) = 13\text{MHz}$). If these results are correct, I would like the authors to add some of these information on the paper so that it is clear for the readers that the dynamically stabilized qubit:

i. is of “low” frequency compare to the usual one, for instance a transmon is usually around 5GHz.

ii. calculated “low” temperature is comparable to a transmon temperature of $\sim 110\text{mK}$ if we take a 5GHz transmon thermally populated a 10%,.

My point here is that I think adding a context to the obtained effective temperature would be beneficial to the paper. Of course feel free to use another angle if you wish, my comparison with a transmon was done to illustrate my point only and not to impose it.

We completely agree with the Reviewer that this comparison is very useful and think that it makes a great addition to the manuscript. As such, we have added several sentences related to both points i) and ii) near the conclusion of the discussion related to the effective temperature.

4. On page 10, the author wrote: “We tomographically reconstruct the steady-state expectation value $\langle \sigma_{x,z} \rangle = \text{Tr}(\rho \sigma_{x,z})$ in the bare qubit eigenbasis as a function of drive parameters and compare with the solutions to Eq. 2 as shown in Fig. 4(a,b)”.

a. The Fig.4.a and Fig.4.b only show σ_x . Please correct the text.

We have corrected this typo and the text now reads “We begin by tomographically reconstructing the steady-state expectation value $\langle \sigma_x \rangle = \text{Tr}(\rho \sigma_x)$ in the bare qubit eigenbasis as a function of drive parameters and compare with the solutions to Eq. 2 as shown in Fig. 4”.

5. On page 11, the author wrote: “As we modify the coupling between the qubit and SAW-phonon bath, we observe excellent agreement between the measured and predicted purity of the resulting qubit state”.

a. I do not understand how the author can discuss the purity of the obtained state with a figure showing the expected value $\langle \sigma_x \rangle$. Maybe this sentence referred to the sup Fig.S4 which would make sense.

b. Please explain.

We agree with the reviewer that this part of the discussion was confusing and that the figure that we need to reference is in fact Fig. S4 in the previous version of the SI. We feel that the Reviewer's comment underscores the importance of this point and that we need to elevate this figure from the SI to the main manuscript body to better support the claims we make in the text. As such, we have removed all discussion of the purity from Fig. 4 and added a new figure (Fig. 5), which focuses solely on the state purity and discusses the purity in reference to this new Figure 5.

6. On page 11, the author wrote: "We find that the results shown in Fig.4.c are also in good agreement with the open quantum systems modeling of the hybrid systems based on Eq. 2".

a. This assertion is currently not supported the paper.

We completely agree with the Reviewer and have added Fig. 5c, which displays both the experimental and modeled values for the steady-state purity as a function of coordinate in the Bloch sphere to support our assertion. Additionally, we have added a discussion of sources of uncertainty leading to differences between measured and modeled values of the tomography components to the main manuscript and the SI.

b. I agree that with the model shown in Fig.3.e as a solid red line (solid red line not explained to the reader) it seems that purity reached for long drive duration is understood. However, this figure only prove it for one set of drive parameters, not for the whole sets shown in Fig.4.c. I would encourage the author to add proof of their claim.

As described above in our response to point a) in the revised version of the manuscript we have promoted the steady-state purity figures from the Supplemental Information to the main text (new Fig. 5(a,b)) so that the reader can see the experimental and simulated steady-state purity as a function of all drive parameters side-by-side.

Additionally we have modified Figure 3e (and its caption) to describe the solid red theory line.

Typo:

1. In the sup page 4, I think the author mean Eq. 2 and not 4.

Correct, we have fixed this typo.

Reviewer #3 (Remarks to the Author):

All points raised by me during the initial review have been adequately addressed. That being said, I would recommend this work for publication.

We again thank the reviewer for their time to help improve our manuscript and for recommending our manuscript for publication.

OTHER MINOR CHANGES:

1. We have corrected the inline equation for the reduced density matrix on page 4 of the SI. In particular the factor of $\frac{1}{2}$ should be in front of all the terms in parentheses.
2. For clarity we have split the “Calibrations and Tomography Data” section of the SI into two separate sections “Calibrations” and “Tomography Data & Analysis”.
3. We have added “titles” to each of the figures in the main text and SI to be in compliance with the convention used in Nature Communications.
4. We have added the data and code availability statements to the manuscript to be in compliance with the conventions used in Nature Communications.

REVIEWERS' COMMENTS

Reviewer #1 (Remarks to the Author):

I thank the authors for their answers to my comments.

I do not find the current draft to be overclaimed.

I believe that the current draft reached sufficient quality to be published.

Response to Reviewers:

Below we have copied the final reviewer comments in red, followed by our response in black.

REVIEWERS' COMMENTS

Reviewer #1 (Remarks to the Author):

I thank the authors for their answers to my comments.

I do not find the current draft to be overclaimed.

I believe that the current draft reached sufficient quality to be published

We thank the reviewer again for their time and constructive input, our paper is the better for it.